# Differentiable Perturb-and-Parse: Semi-Supervised Parsing with a Structured Variational Autoencoder

**Caio Corro**      **Ivan Titov**

ILCC, School of Informatics, University of Edinburgh
ILLC, University of Amsterdam
`c.f.corro@uva.nl`      `ititov@inf.ed.ac.uk`

## Abstract

Human annotation for syntactic parsing is expensive, and large resources are available only for a fraction of languages. A question we ask is whether one can leverage abundant unlabeled texts to improve syntactic parsers, beyond just using the texts to obtain more generalisable lexical features (i.e. beyond word embeddings). To this end, we propose a novel latent-variable generative model for semi-supervised syntactic dependency parsing. As exact inference is intractable, we introduce a differentiable relaxation to obtain approximate samples and compute gradients with respect to the parser parameters. Our method (Differentiable Perturb-and-Parse) relies on differentiable dynamic programming over stochastically perturbed arc weights. We demonstrate effectiveness of our approach with experiments on English, French and Swedish.

## 1 Introduction

A dependency tree is a lightweight syntactic structure exposing (possibly labeled) bi-lexical relations between words (Tesnière, 1959; Kaplan & Bresnan, 1982), see Figure 1. This representation has been widely studied by the NLP community leading to very efficient state-of-the-art parsers (Kiperwasser & Goldberg, 2016; Dozat & Manning, 2017; Ma & Hovy, 2017), motivated by the fact that dependency trees are useful in downstream tasks such as semantic parsing (Reddy et al., 2016; Marcheggiani & Titov, 2017), machine translation (Ding & Palmer, 2005; Bastings et al., 2017), information extraction (Culotta & Sorensen, 2004; Liu et al., 2015), question answering (Cui et al., 2005) and even as a filtering method for constituency parsing (Kong et al., 2015), among others.

Unfortunately, syntactic annotation is a tedious and expensive task, requiring highly-skilled human annotators. Consequently, even though syntactic annotation is now available for many languages, the datasets are often small. For example, 31 languages in the Universal Dependency Treebank,[1] the largest dependency annotation resource, have fewer than 5,000 sentences, including such major languages as Vietnamese and Telugu. This makes the idea of using unlabeled texts as an additional source of supervision especially attractive.

In previous work, before the rise of deep learning, the semi-supervised parsing setting has been mainly tackled with two-step algorithms. On the one hand, feature extraction methods first learn an intermediate representation using an unlabeled dataset which is then used as input to train a supervised parser (Koo et al., 2008; Yu et al., 2008; Chen et al., 2009; Suzuki et al., 2011). On the other hand, the self-training and co-training methods start by learning a supervised parser that is then used to label extra data. Then, the parser is retrained with this additional annotation (Sagae & Tsujii, 2007; Kawahara & Uchimoto, 2008; McClosky et al., 2006). Nowadays, unsupervised feature extraction is achieved in neural parsers by the means of word embeddings (Mikolov et al., 2013; Peters et al., 2018). The natural question to ask is whether one can exploit unlabeled data in neural parsers beyond only inducing generalizable word representations.

---

[1] `http://universaldependencies.org/`

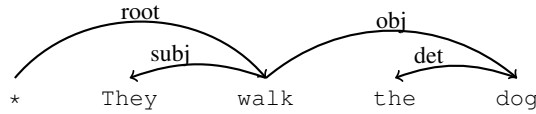

Figure 1: Dependency tree example: each arc represents a labeled relation between the head word (the source of the arc) and the modifier word (the destination of the arc). The first token is a fake root word.

Table 1: Number of labeled and unlabeled instances in each dataset.

|  | **Labeled** | **Unlabeled** |
|---|---|---|
| **English** | 3984 | 35848 |
| **French** | 1476 | 13280 |
| **Swedish** | 4880 | 5331 |

Our method can be regarded as semi-supervised Variational Auto-Encoder (VAE, Kingma et al., 2014). Specifically, we introduce a probabilistic model (Section 3) parametrized with a neural network (Section 4). The model assumes that a sentence is generated conditioned on a latent dependency tree. Dependency parsing corresponds to approximating the posterior distribution over the latent trees within this model, achieved by the encoder component of VAE, see Figure 2a. The parameters of the generative model and the parser (i.e. the encoder) are estimated by maximizing the likelihood of unlabeled sentences. In order to ensure that the latent representation is consistent with treebank annotation, we combine the above objective with maximizing the likelihood of gold parse trees in the labeled data.

Training a VAE via backpropagation requires marginalization over the latent variables, which is intractable for dependency trees. In this case, previous work proposed approximate training methods, mainly differentiable Monte-Carlo estimation (Kingma & Welling, 2013; Rezende et al., 2014) and score function estimation, e.g. REINFORCE (Williams, 1992). However, REINFORCE is known to suffer from high variance (Mnih & Gregor, 2014). Therefore, we propose an approximate differentiable Monte-Carlo approach that we call Differentiable Perturb-and-Parse (Section 5). The key idea is that we can obtain a differentiable relaxation of an approximate sample by (1) perturbing weights of candidate dependencies and (2) performing structured argmax inference with differentiable dynamic programming, relying on the perturbed scores. In this way we bring together ideas of perturb-and-map inference (Papandreou & Yuille, 2011; Maddison et al., 2017) and continuous relaxation for dynamic programming (Mensch & Blondel, 2018). Our model differs from previous works on latent structured models which compute marginal probabilities of individual edges Kim et al. (2017); Liu & Lapata (2018). Instead, we sample a single tree from the distribution that is represented with a soft selection of arcs. Therefore, we preserve higher-order statistics, which can then inform the decoder. Computing marginals would correspond to making strong independence assumptions. We evaluate our semi-supervised parser on English, French and Swedish and show improvement over a comparable supervised baseline (Section 6).

Our main contributions can be summarized as follows: (1) we introduce a variational autoencoder for semi-supervised dependency parsing; (2) we propose the Differentiable Perturb-and-Parse method for its estimation; (3) we demonstrate the effectiveness of the approach on three different languages. In short, we introduce a novel generative model for learning latent syntactic structures.

## 2 DEPENDENCY PARSING

A dependency is a bi-lexical relation between a head word (the source) and a modifier word (the target), see Figure 1. The set of dependencies of a sentence defines a tree-shaped structure.[2] In the parsing problem, we aim to compute the dependency tree of a given sentence.

Formally, we define a sentence as a sequence of tokens (words) from vocabulary $\mathbb{W}$. We assume a one-to-one mapping between $\mathbb{W}$ and integers $1 \dots |\mathbb{W}|$. Therefore, we write a sentence of length $n$ as a vector of integers $\boldsymbol{s}$ of size $n+1$ with $1 \leq s_i \leq |\mathbb{W}|$ and where $s_0$ is a special root symbol. A dependency tree of sentence $\boldsymbol{s}$ is a matrix of booleans $\boldsymbol{T} \in \{0,1\}^{(n+1)\times(n+1)}$ with $T_{h,m} = 1$ meaning that word $s_h$ is the head of word $s_m$ in the dependency tree.

---

[2] Semantic dependencies can have a more complex structure, e.g. words with several heads. However, we focus on syntactic dependencies only.

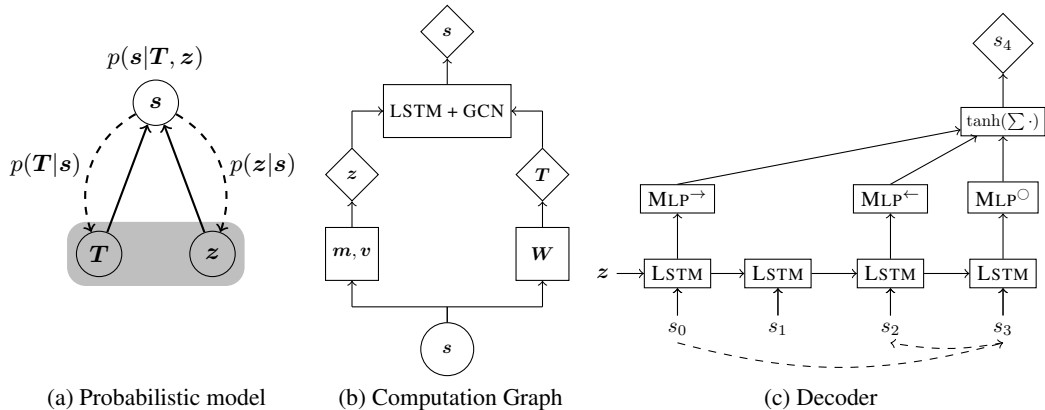

(a) Probabilistic model     (b) Computation Graph     (c) Decoder

Figure 2: **(a)** Illustration of our probabilistic model with random variables $s$, $T$ and $z$ for sentences, dependency trees and sentence embeddings, respectively. The gray area delimits the latent space. Solid arcs denote the generative process, dashed arcs denotes posterior distributions over the latent variables. **(b)** Stochastic computation graph. **(c)** Illustration of the decoder when computing the probability distribution of $s_4$, the word at position 4. Dashed arcs at the bottom represent syntactic dependencies between word at position 4 and previous positions. At each step, the LSTM takes as input an embedding of the previous word ($s_0$ is a special start-of-sentence symbol). Then, the GCN combines different outputs of the LSTM by transforming them with respect to their syntactic relation with the current position. Finally, the probability of $s_4$ is computed via the softmax function.

More specifically, a dependency tree $T$ is the adjacency matrix of a directed graph with $n + 1$ vertices $v_0 \ldots v_n$. A matrix $T$ is a valid dependency tree if and only if this graph is a $v_0$-rooted spanning arborescence,[3] i.e. the graph is connected, each vertex has at most one incoming arc and the only vertex without incoming arc is $v_0$. A dependency tree is projective if and only if, for each arc $v_h \rightarrow v_m$, if $h < m$ (resp. $m < h$) then there exists a path with arcs $T$ from $v_h$ to each vertex $v_k$ such that $h < k < m$ (resp. $m < k < h$). From a linguistic point of view, projective dependency trees combine contiguous phrases (sequence of words) only. Intuitively, this means that we can draw the dependency tree above the sentence without crossing arcs.

Given a sentence $s$, an arc-factored dependency parser computes the dependency tree $T$ which maximizes a weighting function $f(T; W) = \sum_{h,m} T_{h,m} W_{h,m}$, where $W$ is a matrix of dependency (arc) weights. This problem can be solved with a $\mathcal{O}(n^2)$ time complexity (Tarjan, 1977; McDonald et al., 2005). If we restrict $T$ to be a projective dependency tree, then the optimal solution can be computed with a $\mathcal{O}(n^3)$ time complexity using dynamic programming (Eisner, 1996). Restricting the search space to projective trees is appealing for treebanks exhibiting this property (either exactly or approximately): they enforce a structural constraint that can be beneficial for accuracy, especially in a low-resource scenario. Moreover, using a more restricted search space of potential trees may be especially beneficial in a semi-supervised scenario: with a more restricted space a model is less likely to diverge from a treebank grammar and capture non-syntactic phenomena. Finally, Eisner's algorithm (Eisner, 1996) can be described as a deduction system (Pereira & Warren, 1983), a framework that unifies many parsing algorithms. As such, our methodology could be applied to other grammar formalisms. For all these reasons, in this paper, we focus on projective dependency trees only.

## 3 GENERATIVE MODEL

We now turn to the learning problem, i.e. estimation of the matrix $W$. We assume that we have access to a set of i.i.d. labeled sentences $\mathbb{L} = \{\langle s, T \rangle, \ldots \}$ and a set of i.i.d. unlabeled sentences $\mathbb{U} = \{s, \ldots \}$. In order to incorporate unlabeled data in the learning process, we introduce a generative model where the dependency tree is latent (Subsection 3.1). As such, we can maximize the

---

[3] Tree refers to the linguistic structure whereas arborescence refers to the graph structure.

likelihood of observed sentences even if the ground-truth dependency tree is unknown. We learn the parameters of this model using a variational Bayes approximation (Subsection 3.2) augmented with a discriminative objective on labeled data (Subsection 3.3).

## 3.1 GENERATIVE STORY

Under our probabilistic model, a sentence $s$ is generated from a continuous sentence embedding $z$ and with respect to a syntactic structure $T$. We formally define the generative process of a sentence of length $n$ as:

$$T \sim p(T|n) \qquad\qquad z \sim p(z|n) \qquad\qquad s \sim p(s|T, z, n)$$

This Bayesian network is shown in Figure 2a. In order to simplify notation, we omit conditioning on $n$ in the following. $T$ and $z$ are latent variables and $p(s|T, z)$ is the conditional likelihood of observations. We assume that the priors $p(T)$ and $p(z)$ are the uniform distribution over projective trees and the multivariate standard normal distribution, respectively. The true distribution underlying the observed data is unknown, so we have to learn a model $p_\theta(s|T, z)$ parametrized by $\theta$ that best fits the given samples:

$$\theta = \arg\max_\theta \sum_s \log p_\theta(s) \tag{1}$$

Then, the posterior distribution of latent variables $p_\theta(T, z|s)$ models the probability of underlying representations (including dependency trees) with respect to a sentence. This conditional distribution can be written as:

$$p_\theta(T, z|s) = \frac{p_\theta(s|T, z)p(T)p(z)}{p_\theta(s)} \tag{2}$$

In the next subsection, we explain how these two quantities can be estimated from data.

## 3.2 VARIATIONAL AUTO-ENCODERS

Computations in Equation 1 and Equation 2 require marginalization over the latent variables:

$$p_\theta(s) = \sum_T \int p_\theta(s, T, z) \, dz$$

which is intractable in general. We rely on the Variational Auto-Encoder (VAE) framework to tackle this challenge (Kingma & Welling, 2013; Rezende et al., 2014). We introduce a variational distribution $q_\phi(T, z|s)$ which is intended to be similar to $p_\theta(T, z|s)$. More formally, we want $\mathrm{KL}\left[q_\phi(T, z|s)\|p_\theta(T, z|s)\right]$ to be as small as possible, where KL is the Kulback-Leibler (KL) divergence. Then, the following equality holds:

$$\log p_\theta(s) = \mathbb{E}_{q_\phi(T, z|s)}[\log p_\theta(s|T, z)] - \mathrm{KL}[q_\phi(T, z|s)|p(T, z)] + \mathrm{KL}\left[q_\phi(T, z|s)\|p_\theta(T, z|s)\right]$$

where $\log p_\theta(s)$ is called the evidence. The KL divergence is always positive, therefore by removing the last term we have:

$$\log p_\theta(s) \geq \mathbb{E}_{q_\phi(T, z|s)}[\log p_\theta(s|T, z)] - \mathrm{KL}[q_\phi(T, z|s)|p(T, z)] = \tilde{\mathcal{E}}_{\theta,\phi}(s) \tag{3}$$

where the right-hand side is called the Evidence Lower Bound (ELBO). By maximizing the ELBO term, the divergence $\mathrm{KL}\left[q_\phi(T, z|s)\|p_\theta(T, z|s)\right]$ is implicitly minimized. Therefore, we define a surrogate objective, replacing the objective in Equation 1:

$$\theta = \arg\max_\theta \sum_s \max_\phi \tilde{\mathcal{E}}_{\theta,\phi}(s) \tag{4}$$

The ELBO in Equation 4 has two components. First, the KL divergence with the prior, which usually has a closed form solution. For the distribution over dependency trees, it can be computed with the semiring algorithm of Li & Eisner (2009). Second, the non-trivial term $\mathbb{E}_{q_\phi(T, z|s)}[\log p_\theta(s|T, z)]$. During training, Monte-Carlo method provides a tractable and unbiased estimation of the expectation. Note that a single sample from $q_\phi(T, z|s)$ can be understood as encoding the observation

into the latent space, whereas regenerating a sentence from the latent space can be understood as decoding. However, training a VAE requires the sampling process to be differentiable. In the case of the sentence embedding, we follow the usual setting and define $q_\phi(z|s)$ as a diagonal Gaussian: backpropagation through the the sampling process $z \sim q_\phi(z|s)$ can be achieved thanks to the reparametrization trick (Kingma & Welling, 2013; Rezende et al., 2014). Unfortunately, this approach cannot be applied to dependency tree sampling $T \sim q_\phi(T|s)$. We tackle this issue in Section 5.

## 3.3 SEMI-SUPERVISED LEARNING

VAEs are a convenient approach for semi-supervised learning (Kingma et al., 2014) and have been successfully applied in NLP (Kočiský et al., 2016; Xu et al., 2017; Zhou & Neubig, 2017; Yin et al., 2018). In this scenario, we are given the dependency structure of a subset of the observations, i.e. $T$ is an observed variable. Then, the supervised ELBO term is defined as:

$$\bar{\mathcal{E}}_{\theta,\phi}(s, T) = \mathbb{E}_{q_\phi(z|s)}[\log p_\theta(s|T, z)] - \text{KL}[q_\phi(z|s)|p(z)] \tag{5}$$

Note that our end goal is to estimate the posterior ditribution over dependency trees $q_\phi(T|s)$, i.e. the dependency parser, which does not appear in the supervised ELBO. We want to explicitly use the labeled data in order to learn the parameters of this parser. This can be achieved by adding a discriminative training term to the overall loss.[4]

The loss function for training a semi-supervised VAE is:

$$\mathcal{L}_{\theta,\phi}(\mathbb{L}, \mathbb{U}) = - \sum_{s,T \in \mathbb{L}} \log q_\phi(T|s) - \sum_{s,T \in \mathbb{L}} \bar{\mathcal{E}}_{\theta,\phi}(s, T) - \sum_{s \in \mathbb{U}} \tilde{\mathcal{E}}_{\theta,\phi}(s) \tag{6}$$

where the first term is the standard loss for supervised learning of log-linear models (Johnson et al., 1999; Lafferty et al., 2001).

## 4 NEURAL PARAMETRIZATION

In this section, we describe the neural parametrization of the encoder distribution $q_\phi$ (Subsection 4.1) and the decoder distribution $p_\theta$ (Subsection 4.2). A visual representation is given in Figure 2b.

## 4.1 ENCODER

We factorize the encoder as $q_\phi(T, z|s) = q_\phi(T|s)q_\phi(z|s)$. The categorical distribution over dependency trees is parametrized by a log-linear model (Lafferty et al., 2001) where the weight of an arc is given by the neural network of Kiperwasser & Goldberg (2016).The sentence embedding model is specified as a diagonal Gaussian parametrized by a LSTM, similarly to the seq2seq framework (Sutskever et al., 2014; Bowman et al., 2016). That is:

$$W = \text{DEPWEIGHTS}(s) \qquad\qquad m, \log v^2 = \text{EMBPARAMS}(s)$$

$$q_\phi(T|s) = \frac{\exp(\sum_{i,j} W_{i,j} T_{i,j})}{\sum_{T'} \exp(\sum_{i,j} W_{i,j} T'_{i,j})} \qquad\qquad q_\phi(z|s) = \mathcal{N}(z|m, v)$$

where $m$ and $v$ are mean and variance vectors, respectively.[5]

## 4.2 DECODER

We use an autoregressive decoder that combines an LSTM and a Graph Convolutional Network (GCN, Kipf & Welling, 2016; Marcheggiani & Titov, 2017). The LSTM keeps the history of generated words, while the GCN incorporate information about syntactic dependencies.

---

[4] This term can be equivalently regarded as a form of data-dependent prior on the posterior distribution, see Section 3.1.2 in (Kingma et al., 2014).

[5] The covariance matrix can be reduced to a vector as we restrict it to be diagonal.

The hidden state of the LSTM is initialized with latent variable $z$ (the sentence embedding). Then, at each step $1 \leq i \leq n$, an embedding associated with word at position $i - 1$ is fed as input. A special start-of-sentence symbol embedding is used at the first position.

Let $o^i$ be the hidden state of the LSTM at position $i$. The standard seq2seq architecture uses this vector to predict the word at position $i$. Instead, we transform it in order to take into account the syntactic structure described by the latent variable $T$. Due to the autoregressive nature of the decoder, we can only take into account dependencies $T_{h,m}$ such that $h < i$ and $m < i$. Before being fed to the GCN, the output of the LSTM is fed to distinct multi-layer perceptrons[6] that characterize syntactic relations: if $s_h$ is the head of $s_i$, $o^h$ is transformed with $\mathrm{MLP}^{\frown}$, if $s_m$ is a modifier of $s_i$, $o^m$ is transformed with $\mathrm{MLP}^{\frown}$, and lastly $o^i$ is transformed with $\mathrm{MLP}^{\circ}$. Formally, the GCN is defined as follows:

$$g^i = \tanh\left(\mathrm{MLP}^{\circ}(o^i) + \sum_{h=0}^{i-1} T_{h,i} \times \mathrm{MLP}^{\frown}(o^h) + \sum_{m=0}^{i-1} T_{i,m} \times \mathrm{MLP}^{\frown}(o^m)\right)$$

The output vector $g^i$ is then used to estimate the probability of word $s_i$. The neural architecture of the decoder is illustrated on Figure 2c.

## 5 DIFFERENTIABLE PERTURB-AND-PARSE

Encoder-decoder architectures are usually straightforward to optimize with the back-propagation algorithm (Linnainmaa, 1976; LeCun et al., 2012) using any autodiff library. Unfortunately, our VAE contains stochastic nodes that can not be differentiated efficiently as marginalization is too expensive or intractable (see Figure 2b for the list of stochastic nodes in our computation graph). Kingma & Welling (2013) and Rezende et al. (2014) proposed to rely on a Monte-Carlo estimation of the gradient. This approximation is differentiable because the sampling process is moved out of the backpropagation path.[7]

In this section, we introduce our Differentiable Perturb-and-Parse operator to cope with the distribution over dependency trees. Firstly, in Subsection 5.1, we propose an approximate sampling process by computing the best parse tree with respect to independently perturbed arc weights. Secondly, we propose a differentiable surrogate of the parsing algorithm in Subsection 5.2.

### 5.1 PERTURB-AND-PARSE

Sampling from a categorical distributions can be achieved through the Gumbel-Max trick (Gumbel, 1954; Maddison et al., 2014).[8] Unfortunately, this reparametrization is difficult to apply when the discrete variable can take an exponential number of values as in Markov Random Fields (MRF). Papandreou & Yuille (2011) proposed an approximate sampling process: each component is perturbed independently. Then, standard MAP inference algorithm computes the sample. This technique is called perturb-and-map.

Arc-factored dependency parsing can be expressed as a MRF where variable nodes represent arcs, singleton factors weight arcs and a fully connected factor forces the variable assignation to describe a valid dependency tree (Smith & Eisner, 2008). Therefore, we can apply the perturb-and-map method to dependency tree sampling:[9]

$$\begin{aligned} W &= \mathrm{EMBPARAMS}(s) \\ P &\sim \mathcal{G}(0,1) \\ T &= \mathrm{EISNER}(W + P) \end{aligned}$$

where $\mathcal{G}(0,1)$ is the Gumbel distribution, that is sampling matrix $P$ is equivalent to setting $P_{i,j} = -\log(-\log U_{i,j})$ where $U_{i,j} \sim \mathrm{Uniform}(0,1)$.

---

[6] Distinct means that $\mathrm{MLP}^{\frown}$, $\mathrm{MLP}^{\frown}$ and $\mathrm{MLP}^{\circ}$ have different parameters.

[7] We briefly describe the reparametrization trick in Appendix A for self-containedness.

[8] We briefly describe the Gumbel-Max trick in Appendix B for self-containedness.

[9] Alternatively, it is possible to sample from the set of projective dependency trees by running the inside-out algorithm (Eisner, 2016). However, it is then not straightforward to formally derive a path derivative gradient estimation.

**Algorithm 1** This function search the best split point for constructing an element given its span. $b$ is a one-hot vector such that $b_{i-k} = 1$ iff $k$ is the best split position.

1: **function** DEDUCE-URIGHT($i, j, W$)
2:     $s \leftarrow$ null-initialized vec. of size $j - i$
3:     **for** $i \leq k < j$ **do**
4:         $s_{i-k} \leftarrow [i \triangle k]$
                    $+ [k+1 \triangleleft j]$
                    $+ W_{j,i}$
5:     $b \leftarrow$ ONE-HOT-ARGMAX($s$)
6:     BACKPTR$[i \triangleright j] \leftarrow b$
7:     WEIGHT$[i \triangleright j] \leftarrow b^{\top} s$

**Algorithm 2** If item $[i \triangleright j]$ has contributed the optimal objective, this function sets $T_{i,j}$ to 1. Then, it propagates the contribution information to its antecedents.

1: **function** BACKTRACK-URIGHT($i, j, T$)
2:     $T_{i,j} \leftarrow$ CONTRIB$[i \triangleright j]$
3:     $b \leftarrow$ BACKPTR$[i \triangleright j]$
4:     **for** $i \leq k < j$ **do**
5:         CONTRIB$[i \triangle k] \overset{+}{\leftarrow} b_{i-k} T_{i,j}$
6:         CONTRIB$[k+1 \triangleleft j] \overset{+}{\leftarrow} b_{i-k} T_{i,j}$

The (approximate) Monte-Carlo estimation of the expectation in Equation 3 is then defined as:[10]

$$\mathbb{E}_{q_\phi(T|s)}\left[\log p_\theta(s|T)\right] \simeq \log p_\theta(s|\text{EISNER}(W + P))$$

where $\simeq$ denotes a Monte-Carlo estimation of the gradient, $P \sim \mathcal{G}(0, 1)$ is sampled in the last line and EISNER is an algorithm that compute the projective dependency tree with maximum (perturbed) weight (Eisner, 1996). Therefore, the sampling process is outside of the backpropagation path. Unfortunately, the EISNER algorithm is built using ONE-HOT-ARGMAX operations that have ill-defined partial derivatives. We propose a differentiable surrogate in the next section.

## 5.2 DIFFERENTIABLE PARSING ALGORITHM

We now propose a continuous relaxation of the projective dependency parsing algorithm. We start with a brief outline of the algorithm using the parsing-as-deduction formalism, restricting this presentation to the minimum needed to describe our continuous relaxation. We refer the reader to Eisner (1996) for an in-depth presentation.

The parsing-as-deduction formalism provides an unified presentation of many parsing algorithms (Pereira & Warren, 1983; Shieber et al., 1995). In this framework, a parsing algorithm is defined as a deductive system, i.e. as a set of axioms, a goal item and a set of deduction rules. Each deduced item represents a sub-analysis of the input. Regarding implementation, the common way is to rely on dynamic programming: items are deduced in a bottom-up fashion, from smaller sub-analyses to large ones. To this end, intermediate results are stored in a global chart.

For projective dependency parsing, the algorithm builds a chart whose items are of the form $[i \triangleright j]$, $[i \triangleleft j]$, $[i \triangle j]$ and $[i \triangle j]$ that represent sub-analyses from word $i$ to word $j$. An item $[i \triangle j]$ (resp. $[i \triangleright j]$) represents a sub-analysis where every word $s_k, i \leq k \leq j$ is a descendant of $s_i$ and where $s_j$ cannot have any other modifier (resp. can have). The two other types are defined similarly for descendants of word $s_j$. In the first stage of the algorithm, the maximum weight of items are computed (deduced) in a bottom-up fashion. For example, the weight WEIGHT$[i \triangleright j]$ is defined as the maximum of WEIGHT$[i \triangle k]$ + WEIGHT$[k+1 \triangleleft j]$, $\forall k$ s.t. $i \leq k < j$, plus $W_{i,j}$ because $[i \triangleright j]$ assumes a dependency with head $s_i$ and modifier $s_j$. In the second stage, the algorithm retrieves arcs whose scores have contributed to the optimal objective. Part of the pseudo-code for the first and second stages are given in Algorithm 1 and Algorithm 2, respectively. Note that, usually, the second stage is implemented with a linear time complexity but we cannot rely on this optimization for our continuous relaxation.

This algorithm can be thought of as the construction of a computational graph where WEIGHT, BACKPTR and CONTRIB are sets of nodes (variables). This graph includes ONE-HOT-ARGMAX operations that are not differentiable (see line 5 in Algorithm 1). This operation takes as input a vector of weights $v$ of size $k$ and returns a one-hot vector $o$ of the same size with $o_i = 1$ if and only

---

[10] We remove variable $z$ to simplify notation.

if $v_i$ is the element of maximum value:[11]

$$o_i = \mathbb{1}[\forall 1 \leq j \leq k, j \neq i : v_i > v_j]$$

We follow a recent trend (Jang et al., 2017; Maddison et al., 2017; Goyal et al., 2017; 2018) in differentiable approximation of the ONE-HOT-ARGMAX function and replace it with the PEAKED-SOFTMAX operator:

$$o_i = \frac{\exp(1/\tau\ v_i)}{\sum_{1 \leq j \leq k} \exp(1/\tau\ v_j)}$$

where $\tau > 0$ is a temperature hyperparameter controlling the smoothness of the relaxation: when $\tau \to \infty$ the relaxation becomes equivalent to ONE-HOT-ARGMAX. With this update, the parsing algorithm is fully differentiable.[12] Note, however, that outputs are not valid dependency trees anymore. Indeed, then an output matrix $T$ contains continuous values that represent soft selection of arcs. Mensch & Blondel (2018) introduced a alternative but similar approach for tagging with the Viterbi algorithm. We report pseudo-codes for the forward and backward passes of our continuous relaxation of EISNER's algorithm in Appendix F.

## 5.3 DISCUSSION

The fact that $T$ is a soft selection of arcs, and not a combinatorial structure, does not impact the decoder. Indeed, a GCN can be run over weighted graphs, the message passed between nodes is simply multiplied by the continuous weights. This is one of motivations for using GCNs rather than a Recursive LSTMs (Tai et al., 2015) in the decoder. On the one hand, running a GCN with a matrix that represents a soft selection of arcs (i.e. with real values) has the same computational cost than using a standard adjacency matrix (i.e. with binary elements) if we use matrix multiplication on GPU.[13] On the other hand, a recursive network over a soft selection of arcs requires to build a $\mathcal{O}(n^2)$ set of RNN-cells that follow the dynamic programming chart where the possible inputs of a cell are multiplied by their corresponding weight in T, which is expensive and not GPU-friendly.

## 6 EXPERIMENTS

We ran a series of experiments on 3 different languages to test our method for semi-supervised dependency parsing: English, French and Swedish. Details about corpora can be found in Appendix C. The size of each dataset is reported in Table 1. Note that the setting is especially challenging for Swedish: the amount of unlabeled data we use here barely exceeds that of labeled data. The hyperparameters of our network are described in Appendix D. In order to ensure that we do not bias our model for the benefit of the semi-supervised scenario, we use the same parameters as Kiperwasser & Goldberg (2016) for the parser. Also, we did not perform any language-specific parameter selections. This makes us hope that our method can be applied to other languages with little extra effort. We stress that no part-of-speech tags are used as input in any part of our network. For English, the supervised parser took 1.5 hours to train on a NVIDIA Titan X GPU while the semi-supervised parser without sentence embedding, which sees 2 times more instances per epoch, took 3.5 hours to train.

Previous work has shown that learned latent structures tend to differ from linguistic syntactic structures (Kim et al., 2017; Williams et al., 2018). Therefore, we encourage the VAE to rely on latent structures close to the targeted ones by bootstrapping the training procedure with labeled data only. We follow a common practice for VAEs: we experimented with scaling down the KL-divergence of priors (Bowman et al., 2016; Miao et al., 2017; Yin et al., 2018). We use weights 0.01 the KL-divergence with the prior for distributions over sentence embeddings. For dependency trees, we report all experiments with the weight of 0, as removing the term or heavily downweighting it was yielding the best results. As the encoder is bootstrapped with the supervised loss, it is implicitly regularized toward linguistic trees, and the KL term would negate this effect. Intuitively, the KL term favors models which are uncertain on unlabeled examples, which may also be problematic, given that we would expect a strong parser to have sharp posteriors.

---

[11] We assume that there are no ties in the weights, which is very likely to happen because (1) we use randomly initialized deep neural networks and (2) weights are perturbed using random Gumbel noise.

[12] See Appendix E for comparison with semiring parsing (Goodman, 1999).

[13] Sparse matrix multiplication is helpful on CPU, but not always on GPU.

Table 2: **(a)** Parsing results: unlabeled attachment score / labeled attachment score. We also report results with the parser of (Kiperwasser & Goldberg, 2016) which uses a different discriminative loss for supervised training. **(b)** Recall / Precision evaluation with respect to dependency lengths for the supervised parser and the best semi-supervised parser on the English test set. Bold numbers highlight the main differences. **(c)** Recall / Precision evaluation with respect to dependency labels for multi-word expressions (mwe), adverbial modifiers (advmod) and appositional modifiers (appos).

(a) Parsing results

|  | **English** | **French** | **Swedish** |
|---|---|---|---|
| **Supervised** | 88.79 / 84.74 | 84.09 / 77.58 | 86.59 / 78.95 |
| **VAE w. $z$** | 89.39 / 85.44 | 84.43 / 77.89 | 86.92 / 80.01 |
| **VAE w/o $z$** | 89.50 / 85.48 | 84.69 / 78.49 | 86.97 / 79.80 |
| **Kipperwasser & Goldberg** | 89.88 / 86.49 | 84.30 / 77.83 | 86.93 / 80.12 |

(b) Dependency length analysis

| **Distance** | **Supervised Re / Pr** | **Semi-sup. Re / Pr** |
|---|---|---|
| **(to root)** | 93.46 / **89.30** | 93.84 / **92.41** |
| **1** | 95.61 / 94.07 | 95.33 / 94.57 |
| **2** | 93.01 / 90.88 | 92.50 / 92.09 |
| **3 . . . 6** | 85.95 / 88.13 | 87.31 / 87.93 |
| **> 7** | **72.47** / 83.26 | **78.72** / 83.11 |

(c) Dependency label analysis

| **Label** | **Supervised Re / Pr** | **Semi-sup. Re / Pr** |
|---|---|---|
| **mwe** | 75.58 / 81.25 | 90.70 / 84.78 |
| **advmod** | 87.27 / 85.95 | 87.32 / 87.51 |
| **appos** | 77.49 / 80.27 | 81.39 / 81.03 |

## 6.1 PARSING RESULTS

For each dataset, we train under the supervised and the semi-supervised scenario. Moreover, in the semi-supervised setting, we experiment with and without latent sentence embedding $z$. We compare only to the model of Kiperwasser & Goldberg (2016). Recently, even more accurate models have been proposed (e.g., Dozat & Manning, 2017). In principle, the ideas introduced in recent work are mostly orthogonal to our proposal as we can modify our VAE model accordingly. For example, we experimented with using bi-affine attention of Dozat & Manning (2017), though it has not turned out beneficial in our low-resource setting. Comparing to multiple previous parsers would have also required tuning each of them on our dataset, which is infeasible. Therefore, we only report results with a comparable baseline, i.e. trained with a structured hinge loss (Kiperwasser & Goldberg, 2016; Taskar et al., 2005). We did not perform further tuning in order to ensure that our analysis is not skewed toward one setting. Parsing results are summarized in Table 2a.

We observe a score increase in all three languages. Moreover, we observe that VAE performs slightly better without latent sentence embedding. We assume this is due to the fact that dependencies are more useful when no information leaks in the decoder through $z$. Interestingly, we observe an improvement, albeit smaller, even on Swedish, where we used a very limited amount of unlabeled data. We note that training with structured hinge loss gives stronger results than our supervised baseline. In order to maintain the probabilistic interpretation of our model, we did not include a similar term in our model.

We conducted qualitative analyses for English.[14] We report scores with respect to dependency lengths in Table 2b. We observe that the semi-supervised parser tends to correct two kind of errors. Firstly, it makes fewer mistakes on root attachments, i.e. the recall is similar between the two parsers but the precision of the semi-supervised one is higher. We hypothesis that root attachment errors come at a high price in the decoder because there is only a small fraction of the vocabulary that is observed with this syntactic function. Secondly, the semi-supervised parser recovers more long distance relations, i.e. the recall for dependencies with a distance superior or equal to 7 is higher. Intuitively, we assume these dependencies are more useful in the decoder: for short distance dependencies, the LSTM efficiently captures the context of the word to predict, whereas this infor-

---

[14] We used the evaluation script from the SPMRL 2013 shared task: http://www.spmrl.org/spmrl2013-sharedtask.html

mation could be vanishing for long distances, meaning the GCN has more impact on the prediction. We also checked how the scores differ across dependency labels. We report main differences in Tables 2c. The largest improvements are obtained for multi-word expressions: this is particularly interesting because they are known to be challenging in NLP.

## 7 RELATED WORK

Dependency parsing in the low-ressource scenario has been of interest in the NLP community due to the expensive nature of annotation. On the one hand, transfer approaches learn a delexicalized parser for a resource-rich language which is then used to parse a low-resource one (Agić et al., 2016; McDonald et al., 2011). On the other hand, the grammar induction approach learns a dependency parser in an unsupervised manner. Klein & Manning (2004) introduced the first generative model that outperforms the right-branching heuristic in English. Close to our work, Cai et al. (2017) use an auto-encoder setting where the decoder tries to rebuild the source sentence. However, their decoder is unstructured (e.g. it is not auto-regressive).

Variational Auto-Encoders (Kingma & Welling, 2013; Rezende et al., 2014) have been investigated in the semi-supervised settings (Kingma et al., 2014) for NLP. Kočiský et al. (2016) learn a semantic parser where the latent variable is a discrete sequence of symbols. Zhou & Neubig (2017) successfully applied the variational method to semi-supervised morphological re-inflection where discrete latent variables represent linguistic features (e.g. tense, part-of-speech tag). Yin et al. (2018) proposed a semi-supervised semantic parser. Similarly to our model, they rely on a structured latent variable. However, all of these systems use either categorical random variables or the REINFORCE score estimator. To the best of our knowledge, no previous work used continuous relaxation of a dynamic programming latent variable in the VAE setting.

The main challenge is backpropagation through discrete random variables. Maddison et al. (2017) and Jang et al. (2017) first introduced the Gumbel-Softmax operator for the categorical distribution. There are two issues regarding more complex discrete distributions. Firstly, one have to build a reparametrization of the the sampling process. Papandreou & Yuille (2011) showed that low-order perturbations provide samples of good qualities for graphical models. Secondly, one have to build a good differentiable surrogate to the structured $\arg\max$ operator. Early work replaced the structured $\arg\max$ with structured attention (Kim et al., 2017). However, computing the marginals over the parse forest is sensitive to numerical stability outside specific cases like non-projective dependency parsing (Liu & Lapata, 2018; Tran & Bisk, 2018). Mensch & Blondel (2018) proposed a stable algorithm based on dynamic program smoothing. Our approach is highly related but we describe a continuous relaxation using the parsing-as-deduction formalism. Peng et al. (2018) propose to replace the true gradient with a proxy that tries to satisfy constraints on a $\arg\max$ operator via a projection. However, their approach is computationally expensive, so they remove the tree constraint on dependencies during backpropagation. A parallel line of work focuses on sparse structures that are differentiable (Martins & Astudillo, 2016; Niculae et al., 2018).

## 8 CONCLUSIONS

We presented a novel generative learning approach for semi-supervised dependency parsing. We model the dependency structure of a sentence as a latent variable and build a VAE. We hope to motivate investigation of latent syntactic structures via differentiable dynamic programming in neural networks. Future work includes research for an informative prior for the dependency tree distribution, for example by introducing linguistic knowledge (Naseem et al., 2010; Noji et al., 2016) or with an adversarial training criterion Makhzani et al. (2016). This work could also be extended to the unsupervised scenario.

## ACKNOWLEDGMENTS

We thank Diego Marcheggiani, Wilker Ferreira Aziz and Serhii Havrylov for their comments and suggestions. We thank the anonymous reviewers for their comments. The project was supported by the Dutch National Science Foundation (NWO VIDI 639.022.518) and European Research Council (ERC Starting Grant BroadSem 678254).

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

## A  REPARAMETRIZATION TRICK

Sampling from a diagonal Gaussian random variable with mean vector $m$ and variance vector $v$ can be re-expressed as:

$$e \sim \mathcal{N}(0, 1)$$
$$z = m + v \times e$$

where $\boldsymbol{z}$ is the sample. As such, $\boldsymbol{e} \sim \mathcal{N}(0, 1)$ is an input of the neural network for which we do not need to compute partial derivatives. This technique is called the reparametrization trick (Kingma & Welling, 2013; Rezende et al., 2014).

## B  GUMBEL-MAX TRICK

Sampling from a categorical distributions can be achieved through the Gumbel-Max trick (Gumbel, 1954; Maddison et al., 2014). Randomly generated Gumbel noise is added to the log-probability of every element of the sample space. Then, the sample is simply the element with maximum *perturbed* log-probability. Let $\boldsymbol{d} \in \triangle^k$ be a random variable taking values in the corner of the unit-simplex of dimension $k$ with probability:

$$p(\boldsymbol{d} \in \triangle^k) = \frac{\exp(\boldsymbol{w}^\top \boldsymbol{d})}{\sum_{\boldsymbol{d}' \in \triangle^k} \exp(\boldsymbol{w}^\top \boldsymbol{d}')}$$

where $\boldsymbol{w}$ is a vector of weights. Sampling $\boldsymbol{d} \sim p(\boldsymbol{d})$ can be re-expressed as follows:

$$\boldsymbol{g} \sim \mathcal{G}(0, 1)$$
$$\boldsymbol{d} = \arg\max_{\boldsymbol{d} \in \triangle^k} (\boldsymbol{w} + \boldsymbol{g})^\top \boldsymbol{d}$$

where $\mathcal{G}(0, 1)$ is the Gumbel distribution. Sampling $\boldsymbol{g} \sim \mathcal{G}(0, 1)$ is equivalent to setting $g_i = -\log(-\log u_i))$ where $u_i \sim \mathrm{Uniform}(0, 1)$. If $\boldsymbol{w}$ is computed by a neural network, the sampling process is outside the backpropagation path.

## C  CORPORA

**English** We use the Stanford Dependency conversion (De Marneffe & Manning, 2008) of the Penn Treebank (Marcus et al., 1993) with the usual section split: 02-21 for training, 22 for development and 23 for testing. In order to simulate our framework under a low-resource setting, the annotation is kept for $10\%$ of the training set only: a labeled sentence is the sentence which has an index (in the training set) modulo 10 equal to zero.

**French** We use a similar setting with the French Treebank version distributed for the SPMRL 2013 shared task and the provided train/dev/test split (Abeillé et al., 2000; Seddah et al., 2013).

**Swedish** We use the Talbanken dataset (Nivre et al., 2006) which contains two written text parts: the professional prose part (P) and the high school students' essays part (G). We drop the annotation of (G) in order to use this section as unlabeled data. We split the (P) section in labeled train/dev/test using a pseudo-randomized scheme. We follow the splitting scheme of Hall et al. (2006) but fix section 9 as development instead of $k$-fold cross-validation. Sentence $i$ is allocated to section $i$ mod 10. Then, section 1-8 are used for training, section 9 for dev and section 0 for test.

## D  HYPER-PARAMETERS

**Encoder: word embeddings** We concatenate trainable word embeddings of size 100 with external word embeddings.[15] We use the word-dropout settings of Kiperwasser & Goldberg (2016). For English, external embeddings are pre-trained with the structured skip n-gram objective (Ling et al., 2015).[16] For French and Swedish, we use the Polyglot embeddings (Al-Rfou et al., 2013).[17] We stress out that no part-of-speech tag is used as input in any part of our network.

**Encoder: dependency parser** The dependency parser is built upon a two-stack BiLSTM with a hidden layer size of 125 (i.e. the output at each position is of size 250). Each dependency is then weighted using a single-layer perceptron with a tanh activation function. Arc label prediction rely

---

[15] The external embeddings are not updated when training our network.

[16] We use the pre-trained embeddings distributed by Dyer et al. (2015).

[17] We use the pre-trained embeddings distributed at https://sites.google.com/site/rmyeid/projects/polyglot

on a similar setting, we refer to the reader to Kiperwasser & Goldberg (2016) for more information about the parser's architecture.

**Encoder: sentence embedding** The sentence is encoded into a fixed size vector with a simple left-to-right LSTM with an hidden size of 100. The hidden layer at the last position of the sentence is then fed to two distinct single-layer perceptrons, with an output size of 100 followed by a piecewise $\tanh$ activation function, that computes means and standard deviations of the diagonal Gaussian distribution.

**Decoder** The decoder use fixed pre-trained embeddings only. The recurrent layer of the decoder is a LSTM with an hidden layer size of 100. $\text{MLP}^{\frown}$, $\text{MLP}^{\frown}$ and $\text{MLP}^{\circ}$ are all single-layer perceptrons with an output size of 100 and without activation function.

**Training** We encourage the VAE to rely on latent structures close to the targeted ones by bootstrapping the training procedure with labeled data only. In the first two epochs, we train the network with the discriminative loss only. Then, for the next two epochs, we add the supervised ELBO term (Equation 5). Finally, after the 6th epoch, we also add the unsupervised ELBO term (Equation 3). We train our network using stochastic gradient descent for 30 epochs using Adadelta (Zeiler, 2012) with default parameters as provided by the Dynet library (Neubig et al., 2017). In the semi-supervised scenario, we alternate between labeled and unlabeled instances. The temperature of the PEAKED-SOFTMAX operator is fixed to $\tau = 1$.

## E    COMPARISON WITH SEMIRING PARSING

Dynamic programs for parsing have been studied as abstract algorithms that can be instantiated with different semirings (Goodman, 1999). For example, computing the weight of the best parse relies on the $\langle \mathbb{R}, \max, + \rangle$ semiring. This semiring can be augmented with set-valued operations to retrieve the best derivation. However, a straightforward implementation would have a $\mathcal{O}(n^5)$ space complexity: for each item in the chart, we also need to store the set of arcs. Under this formalism, the backpointer trick is a method to implicitly constructs these sets and maintain the optimal $\mathcal{O}(n^3)$ complexity. Our continuous relaxation replaces the $\max$ operator with a smooth surrogate and the set values with a soft-selection of sets. Unfortunately, $\langle \mathbb{R}, \text{PEAKED-SOFTMAX} \rangle$ is not a commutative monoid, therefore the semiring analogy is not transposable.

## F    DIFFERENTIABLE DYNAMIC PROGRAMMING FOR PROJECTIVE DEPENDENCY PARSING

We describe how we can embed a continuous relaxation of projective dependency parsing as a node in a neural network. During the forward pass, we are given arc weights $\boldsymbol{W}$ and we compute the relaxed projective dependency tree $\boldsymbol{T}$ that maximize the arc-factored weight $\sum_{h,m} T_{h,m} \times W_{h,m}$. Each output variable $T_{h,m} \in [0, 1]$ is a soft selection of dependency with head-word $s_h$ and modifier $s_m$. During back-propagation, we are given partial derivatives of the loss with respect to each arc and we compute the ones with respect to arc weights:

$$\frac{\partial \mathcal{L}}{\partial W_{h,m}} = \sum_{i,j} \frac{\partial \mathcal{L}}{\partial T_{i,j}} \frac{\partial T_{i,j}}{\partial W_{h,m}}$$

Note that the Jacobian matrix has $\mathcal{O}(n^4)$ values but we do need to explicitly compute it. The space and time complexity of the forward and backward passes are both cubic, similar to Eisner's algorithm.

### F.1    FORWARD PASS

The forward pass is a two step algorithm:

1. First, we compute the cumulative weight of each item and store soft backpointers to keep track of contribution of antecedents. This step is commonly called to inside algorithm.
2. Then, we compute the contribution of each arc thanks to the backpointers. This step is somewhat similar to the $\arg\max$ reconstruction algorithm.

The outline of the algorithm is given in Algorithm 3.

The inside algorithm computes the following variables:

- $a[i \triangleright j][k]$ is the weight of item $[i \triangleright j]$ if we split its antecedent at $k$.

- $b[i \triangleright j][k]$ is the soft backpointer to antecedents of item $[i \triangleright j]$ with split at $k$.

- $c[i \triangleright j]$ is the cumulative weight of item $[i \triangleright j]$.

and similarly for the other chart values. The algorithm is given in Algorithm 5.

The backpointer reconstruction algorithm compute the contribution of each arc. We follow backpointers in reverse order in order to compute the contribution of each item $\tilde{c}[i \triangleright j]$. The algorithm is given in Algorithm 6.

### F.2 BACKWARD PASS

During the backward pass, we compute the partial derivatives of variables using the chain rule, i.e. in the reverse order of their creation: we first run backpropagation through the backpointer reconstruction algorithm and then through the inside algorithm (see Algorithm 4). Given the partial derivatives in Figure 3, backpropagation through the backpointer reconstruction algorithm is straighforward to compute, see Algorithm 7. Partial derivatives of the inside algorithm's variables are given in Figure 4.

---

**Algorithm 3** Forward algorithm

> **function** RELAXED-EISNER()
>> INSIDE( )
>> BACKPTR()
>>
>> **for** $i = 0 \ldots n$ **do**
>>> **for** $j = 1 \ldots n$ **do**
>>>> **if** $i < j$ **then**
>>>>> $T_{i,j} \leftarrow \tilde{c}[i \triangleright j]$
>>>> **else if** $j < i$ **then**
>>>>> $T_{i,j} \leftarrow \tilde{c}[i \triangleleft j]$

---

**Algorithm 4** Backward algorithm

> **function** BACKPROP-RELAXED-EISNER()
>> BACKPROP-BACKPTR()
>> BACKPROP-INSIDE()
>>
>> **for** $i = 0 \ldots n$ **do**
>>> **for** $j = 1 \ldots n$ **do**
>>>> **if** $i < j$ **then**
>>>>> $\frac{\partial \mathcal{L}}{\partial W_{i,j}} \leftarrow \frac{\partial \mathcal{L}}{\partial c[i \triangleright j]}$
>>>> **else if** $j < i$ **then**
>>>>> $\frac{\partial \mathcal{L}}{\partial W_{i,j}} \leftarrow \frac{\partial \mathcal{L}}{\partial c[j \triangleleft i]}$

---

---

**Algorithm 5** Inside algorithm - Forward pass

---

**function** INSIDE($n$)
    **for** $i \leftarrow 0 \ldots n$ **do**
        $c[i \vartriangleright i] \leftarrow 0, c[i \vartriangleleft i] \leftarrow 0, c[i \triangleright i] \leftarrow 0, c[i \triangleleft i] \leftarrow 0$
    **for** $l \leftarrow 1 \ldots n$ **do**
        **for** $i \leftarrow 0 \ldots n - l$ **do**
            $j \leftarrow i + l$

            **for** $k = i \ldots j - 1$ **do**
                $a[i \triangleright j][k] \leftarrow c[i \vartriangleright k] + c[k + 1 \vartriangleleft j]$
            $b[i \triangleright j] \leftarrow \mathrm{softmax}(a[i \triangleright j])$
            $c[i \triangleright j] \leftarrow W_{i,j} + \sum_{k=i \ldots j-1} b[i \triangleright j][k] \times a[i \triangleright j][k]$

            **for** $k = i \ldots j - 1$ **do**
                $a[i \triangleleft j][k] \leftarrow c[i \vartriangleright k] + c[k + 1 \vartriangleleft j]$
            $b[i \triangleleft j] \leftarrow \mathrm{softmax}(a[i \triangleleft j])$
            $c[i \triangleleft j] \leftarrow W_{j,i} + \sum_{k=i \ldots j-1} b[i \triangleleft j][k] \times a[i \triangleleft j][k]$

            **for** $k = i + 1 \ldots j$ **do**
                $a[i \vartriangleright j][k] \leftarrow c[i \triangleright k] + c[k \vartriangleright j]$
            $b[i \vartriangleright j] \leftarrow \mathrm{softmax}(a[i \vartriangleright j])$
            $c[i \vartriangleright j] \leftarrow \sum_{k=i+1 \ldots j} b[i \vartriangleright j][k] \times a[i \vartriangleright j][k]$

            **for** $k = i \ldots j - 1$ **do**
                $a[i \vartriangleleft j][k] \leftarrow c[i \vartriangleleft k] + c[k \triangleleft j]$
            $b[i \vartriangleleft j] \leftarrow \mathrm{softmax}(a[i \vartriangleleft j])$
            $c[i \vartriangleleft j] \leftarrow \sum_{k=i \ldots j-1} b[i \vartriangleleft j][k] \times a[i \vartriangleleft j][k]$

---

---

**Algorithm 6** Backpointer reconstruction algorithm - Forward pass

---

**function** BACKPTR()
    **for** $i = 0 \ldots n$ **do**
        **for** $j = i \ldots n$ **do**
            $\tilde{c}[i \triangleright j] \leftarrow 0, \tilde{c}[i \triangleleft j] \leftarrow 0, \tilde{c}[i \triangleright\!\!\!\! j] \leftarrow 0, \tilde{c}[i \triangleleft\!\!\!\! j] \leftarrow 0$
    $\tilde{c}[0 \triangleright\!\!\!\! n] \leftarrow 1$

    **for** $l = n \ldots 1$ **do**
        **for** $i = 0 \ldots n - l$ **do**
            $j \leftarrow i + l$

            **for** $k = i + 1 \ldots j$ **do**
                $\tilde{c}[i \triangleright k] \overset{+}{\leftarrow} \tilde{c}[i \triangleright\!\!\!\! j] \times b[i \triangleright\!\!\!\! j][k]$
                $\tilde{c}[k \triangleright\!\!\!\! j] \overset{+}{\leftarrow} \tilde{c}[i \triangleright\!\!\!\! j] \times b[i \triangleright\!\!\!\! j][k]$

            **for** $k = i \ldots j - 1$ **do**
                $\tilde{c}[i \triangleleft\!\!\!\! k] \overset{+}{\leftarrow} \tilde{c}[i \triangleleft\!\!\!\! j] \times b[i \triangleleft\!\!\!\! j][k]$
                $\tilde{c}[k \triangleleft j] \overset{+}{\leftarrow} \tilde{c}[i \triangleleft\!\!\!\! j] \times b[i \triangleleft\!\!\!\! j][k]$

            **for** $k = i \ldots j - 1$ **do**
                $\tilde{c}[i \triangleright\!\!\!\! k] \overset{+}{\leftarrow} \tilde{c}[i \triangleright j] \times b[i \triangleright j][k]$
                $\tilde{c}[k + 1 \triangleleft\!\!\!\! j] \overset{+}{\leftarrow} \tilde{c}[i \triangleright j] \times b[i \triangleright j][k]$

            **for** $k = i \ldots j - 1$ **do**
                $\tilde{c}[i \triangleright\!\!\!\! k] \overset{+}{\leftarrow} \tilde{c}[i \triangleleft j] \times b[i \triangleleft j][k]$
                $\tilde{c}[k + 1 \triangleleft\!\!\!\! j] \overset{+}{\leftarrow} \tilde{c}[i \triangleleft j] \times b[i \triangleleft j][k]$

---

$$\forall i < k \le j : \qquad \frac{\partial \tilde{c}[i \triangleright k]}{\partial \tilde{c}[i \triangleright\!\!\!\! j]} = b[i \triangleright\!\!\!\! j][k] \qquad \forall i < k \le j : \qquad \frac{\partial \tilde{c}[i \triangleright k]}{\partial b[i \triangleright\!\!\!\! j][k]} = \tilde{c}[i \triangleright\!\!\!\! j]$$

$$\forall i < k \le j : \qquad \frac{\partial \tilde{c}[k \triangleright\!\!\!\! j]}{\partial \tilde{c}[i \triangleright\!\!\!\! j]} = b[i \triangleright\!\!\!\! j][k] \qquad \forall i < k \le j : \qquad \frac{\partial \tilde{c}[k \triangleright\!\!\!\! j]}{\partial b[i \triangleright\!\!\!\! j][k]} = \tilde{c}[i \triangleright\!\!\!\! j]$$

$$\forall i \le k < j : \qquad \frac{\partial \tilde{c}[i \triangleleft\!\!\!\! k]}{\partial \tilde{c}[i \triangleleft\!\!\!\! j]} = b[i \triangleleft\!\!\!\! j][k] \qquad \forall i \le k < j : \qquad \frac{\partial \tilde{c}[i \triangleleft\!\!\!\! k]}{\partial b[i \triangleleft\!\!\!\! j][k]} = \tilde{c}[i \triangleleft\!\!\!\! j]$$

$$\forall i \le k < j : \qquad \frac{\partial \tilde{c}[k \triangleleft j]}{\partial \tilde{c}[i \triangleleft\!\!\!\! j]} = b[i \triangleleft\!\!\!\! j][k] \qquad \forall i \le k < j : \qquad \frac{\partial \tilde{c}[k \triangleleft j]}{\partial b[i \triangleleft\!\!\!\! j][k]} = \tilde{c}[i \triangleleft\!\!\!\! j]$$

$$\forall i \le k < j : \qquad \frac{\partial \tilde{c}[i \triangleright\!\!\!\! k]}{\partial \tilde{c}[i \triangleright j]} = b[i \triangleright j][k] \qquad \forall i \le k < j : \qquad \frac{\partial \tilde{c}[i \triangleright\!\!\!\! k]}{\partial b[i \triangleright j][k]} = \tilde{c}[i \triangleright j]$$

$$\forall i \le k < j : \qquad \frac{\partial \tilde{c}[k + 1 \triangleleft\!\!\!\! j]}{\partial \tilde{c}[i \triangleright j]} = b[i \triangleright j][k] \qquad \forall i \le k < j : \qquad \frac{\partial \tilde{c}[k + 1 \triangleleft\!\!\!\! j]}{\partial b[i \triangleright j][k]} = \tilde{c}[i \triangleright j]$$

$$\forall i \le k < j : \qquad \frac{\partial \tilde{c}[i \triangleright\!\!\!\! k]}{\partial \tilde{c}[i \triangleleft j]} = b[i \triangleleft j][k] \qquad \forall i \le k < j : \qquad \frac{\partial \tilde{c}[i \triangleright\!\!\!\! k]}{\partial b[i \triangleleft j][k]} = \tilde{c}[i \triangleleft j]$$

$$\forall i \le k < j : \qquad \frac{\partial \tilde{c}[k + 1 \triangleleft\!\!\!\! j]}{\partial \tilde{c}[i \triangleleft j]} = b[i \triangleleft j][k] \qquad \forall i \le k < j : \qquad \frac{\partial \tilde{c}[k + 1 \triangleleft\!\!\!\! j]}{\partial b[i \triangleleft j][k]} = \tilde{c}[i \triangleleft j]$$

Figure 3: Partial derivatives of the backpointer reconstruction algorithm

---

**Algorithm 7** Backpointer reconstruction algorithm - Backward pass

---

**function** BACKPROP-BACKPTR($n$)
    **for** $l = 1 \dots n$ **do**
        **for** $i = 0 \dots n - l$ **do**
            $j \leftarrow i + l$

$$\frac{\partial \mathcal{L}}{\partial \tilde{c}[i \triangleleft j]} \leftarrow 0, \ \frac{\partial \mathcal{L}}{\partial \tilde{c}[i \triangleright j]} \leftarrow 0, \ \frac{\partial \mathcal{L}}{\partial \tilde{c}[i \triangle j]} \leftarrow 0, \ \frac{\partial \mathcal{L}}{\partial \tilde{c}[i \triangle j]} \leftarrow 0$$

            **for** $k = i \dots j - 1$ **do**

$$\frac{\partial \mathcal{L}}{\partial \tilde{c}[i \triangleleft j]} \overset{+}{\leftarrow} \frac{\partial \mathcal{L}}{\partial \tilde{c}[i \triangle k]} b[i \triangleleft j][k] + \frac{\partial \mathcal{L}}{\partial \tilde{c}[k+1 \triangle j]} b[i \triangleleft j][k]$$

$$\frac{\partial \mathcal{L}}{\partial b[i \triangleleft j][k]} \leftarrow \frac{\partial \mathcal{L}}{\partial \tilde{c}[i \triangle k]} \tilde{c}[i \triangleleft j] + \frac{\partial \mathcal{L}}{\partial \tilde{c}[k+1 \triangle j]} \tilde{c}[i \triangleleft j]$$

            **for** $k = i \dots j - 1$ **do**

$$\frac{\partial \mathcal{L}}{\partial \tilde{c}[i \triangleright j]} \overset{+}{\leftarrow} \frac{\partial \mathcal{L}}{\partial \tilde{c}[i \triangle k]} b[i \triangleright j][k] + \frac{\partial \mathcal{L}}{\partial \tilde{c}[k+1 \triangle j]} b[i \triangleright j][k]$$

$$\frac{\partial \mathcal{L}}{\partial b[i \triangleright j][k]} \leftarrow \frac{\partial \mathcal{L}}{\partial \tilde{c}[i \triangle k]} \tilde{c}[i \triangleright j] + \frac{\partial \mathcal{L}}{\partial \tilde{c}[k+1 \triangle j]} \tilde{c}[i \triangleright j]$$

            **for** $= i \dots j - 1$ **do**

$$\frac{\partial \mathcal{L}}{\partial \tilde{c}[i \triangle j]} \overset{+}{\leftarrow} \frac{\partial \mathcal{L}}{\partial \tilde{c}[i \triangle k]} b[i \triangle j][k] + \frac{\partial \mathcal{L}}{\partial \tilde{c}[k \triangleleft j]} b[i \triangle j][k]$$

$$\frac{\partial \mathcal{L}}{\partial b[i \triangle j][k]} \leftarrow \frac{\partial \mathcal{L}}{\partial \tilde{c}[i \triangle k]} \tilde{c}[i \triangle j] + \frac{\partial \mathcal{L}}{\partial \tilde{c}[k \triangleleft j]} \tilde{c}[i \triangle j]$$

            **for** $= i + 1 \dots j$ **do**

$$\frac{\partial \mathcal{L}}{\partial \tilde{c}[i \triangle j]} \overset{+}{\leftarrow} \frac{\partial \mathcal{L}}{\partial \tilde{c}[i \triangle k]} b[i \triangle j][k] + \frac{\partial \mathcal{L}}{\partial \tilde{c}[k \triangle j]} b[i \triangle j][k]$$

$$\frac{\partial \mathcal{L}}{\partial b[i \triangle j][k]} \leftarrow \frac{\partial \mathcal{L}}{\partial \tilde{c}[i \triangle k]} \tilde{c}[i \triangle j] + \frac{\partial \mathcal{L}}{\partial \tilde{c}[k \triangle j]} \tilde{c}[i \triangle j]$$

---

$$\forall i \leq k < j: \qquad \frac{\partial a[i \rhd j][k]}{\partial c[i \lsquigarrow k]} = 1 \qquad\qquad \forall i \leq k < j: \qquad \frac{\partial a[i \rhd j][k]}{\partial c[k+1 \vartriangleleft j]} = 1$$

$$\forall i \leq k < j: \qquad \frac{\partial a[i \vartriangleleft j][k]}{\partial c[i \lsquigarrow k]} = 1 \qquad\qquad \forall i \leq k < j: \qquad \frac{\partial a[i \vartriangleleft j][k]}{\partial c[k+1 \vartriangleleft j]} = 1$$

$$\forall i < k \leq j: \qquad \frac{\partial a[i \rsquigarrow j][k]}{\partial c[i \rhd k]} = 1 \qquad\qquad \forall i < k \leq j: \qquad \frac{\partial a[i \rsquigarrow j][k]}{\partial c[k \rsquigarrow j]} = 1$$

$$\forall i \leq k < j: \qquad \frac{\partial a[i \vartriangleleft j][k]}{\partial c[i \vartriangleleft k]} = 1 \qquad\qquad \forall i \leq k < j: \qquad \frac{\partial a[i \vartriangleleft j][k]}{\partial c[k \vartriangleleft j]} = 1$$

$$\forall i \leq k, k' < j: \qquad \frac{\partial b[i \rhd j][k]}{\partial a[i \rhd j][k']} = b[i \rhd j][k](\mathbb{1}[k = k'] - b[i \rhd j][k'])$$

$$\forall i \leq k, k' < j: \qquad \frac{\partial b[i \vartriangleleft j][k]}{\partial a[i \vartriangleleft j][k']} = b[i \vartriangleleft j][k](\mathbb{1}[k = k'] - b[i \vartriangleleft j][k'])$$

$$\forall i < k \leq j: \qquad \frac{\partial b[i \rsquigarrow j][k]}{\partial a[i \rsquigarrow j][k']} = b[i \rsquigarrow j][k](\mathbb{1}[k = k'] - b[i \rsquigarrow j][k'])$$

$$\forall i \leq k < j: \qquad \frac{\partial b[i \vartriangleleft j][k]}{\partial a[i \vartriangleleft j][k']} = b[i \vartriangleleft j][k](\mathbb{1}[k = k'] - b[i \vartriangleleft j][k'])$$

$$\forall i \leq k < j: \qquad \frac{\partial c[i \rhd j]}{\partial b[i \rhd j][k]} = a[i \rhd j][k] \qquad \forall i \leq k < j: \qquad \frac{\partial c[i \rhd j]}{\partial a[i \rhd j][k]} = b[i \rhd j][k]$$

$$\forall i \leq k < j: \qquad \frac{\partial c[i \vartriangleleft j]}{\partial b[i \vartriangleleft j][k]} = a[i \vartriangleleft j][k] \qquad \forall i \leq k < j: \qquad \frac{\partial c[i \vartriangleleft j]}{\partial a[i \vartriangleleft j][k]} = b[i \vartriangleleft j][k]$$

$$\forall i < k \leq j: \qquad \frac{\partial c[i \rsquigarrow j]}{\partial b[i \rsquigarrow j][k]} = a[i \rsquigarrow j][k] \qquad \forall i < k \leq j: \qquad \frac{\partial c[i \rsquigarrow j]}{\partial a[i \rsquigarrow j][k]} = b[i \rsquigarrow j][k]$$

$$\forall i \leq k < j: \qquad \frac{\partial c[i \vartriangleleft j]}{\partial b[i \vartriangleleft j][k]} = a[i \vartriangleleft j][k] \qquad \forall i \leq k < j: \qquad \frac{\partial c[i \vartriangleleft j]}{\partial a[i \vartriangleleft j][k]} = b[i \vartriangleleft j][k]$$

$$\forall i < j: \qquad \frac{\partial c[i \rhd j]}{\partial W_{i,j}} = 1 \qquad\qquad \forall i < j: \qquad \frac{\partial c[i \vartriangleleft j]}{\partial W_{j,i}} = 1$$

Figure 4: Partial derivatives of the inside algorithm

---

**Algorithm 8** Inside algorithm - Backward pass

---

**function** BACKPROP-INSIDE($n$)

    **for** $i = 0 \ldots n$ **do**

        **for** $j = i \ldots n$ **do**

            $\frac{\partial \mathcal{L}}{\partial c[i \triangle j]} \leftarrow 0, \frac{\partial \mathcal{L}}{\partial c[i \triangle j]} \leftarrow 0, \frac{\partial \mathcal{L}}{\partial c[i \triangle j]} \leftarrow 0, \frac{\partial \mathcal{L}}{\partial c[i \triangle j]} \leftarrow 0$

    **for** $l = n \ldots 1$ **do**                             $\triangleright$ Backpropagation through the "inside" algorithm

        **for** $i = 0 \ldots n - l$ **do**

            $j \leftarrow i + l$

            **for** $k = i \ldots j - 1$ **do**

                $\frac{\partial \mathcal{L}}{\partial b[i \triangle j][k]} \leftarrow \frac{\partial \mathcal{L}}{\partial c[i \triangle j]} a[i \triangle j][k]$

                $\frac{\partial \mathcal{L}}{\partial a[i \triangle j][k]} \leftarrow \frac{\partial \mathcal{L}}{\partial c[i \triangle j]} b[i \triangle j][k]$

            $s = \sum_{k=i \ldots j-1} \frac{\partial \mathcal{L}}{\partial b[i \triangle j][k]} b[i \triangle j][k]$    $\triangleright$ Backpropagate through the softmax function

            **for** $k = i \ldots j - 1$ **do**

                $\frac{\partial \mathcal{L}}{\partial a[i \triangle j][k]} \leftarrow b[i \triangle j][k] \left( \frac{\partial \mathcal{L}}{\partial b[i \triangle j][k]} - s \right)$

            **for** $k = i \ldots j - 1$ **do**

                $\frac{\partial \mathcal{L}}{\partial c[i \triangle k]} \overset{+}{\leftarrow} \frac{\partial \mathcal{L}}{\partial a[i \triangle j][k]}$

                $\frac{\partial \mathcal{L}}{\partial c[k \triangle j]} \overset{+}{\leftarrow} \frac{\partial \mathcal{L}}{\partial a[i \triangle j][k]}$

            **for** $k = i + 1 \ldots j$ **do**

                $\frac{\partial \mathcal{L}}{\partial b[i \triangle j][k]} \leftarrow \frac{\partial \mathcal{L}}{\partial c[i \triangle j]} a[i \triangle j][k]$

                $\frac{\partial \mathcal{L}}{a[i \triangle j][k]} \leftarrow \frac{\partial \mathcal{L}}{\partial c[i \triangle j]} \partial b[i \triangle j][k]$

            $s = \sum_{k=i+1 \ldots j} \frac{\partial \mathcal{L}}{\partial b[i \triangle j][k]} b[i \triangle j][k]$    $\triangleright$ Backpropagate through the softmax function

            **for** $k = i + 1 \ldots j$ **do**

                $\frac{\partial \mathcal{L}}{\partial a[i \triangle j][k]} \leftarrow b[i \triangle j][k] \left( \frac{\partial \mathcal{L}}{\partial b[i \triangle j][k]} - s \right)$

            **for** $k = i + 1 \ldots j$ **do**

                $\frac{\partial \mathcal{L}}{\partial c[i \triangle k]} \overset{+}{\leftarrow} \frac{\partial \mathcal{L}}{\partial a[i \triangle j][k]}$

                $\frac{\partial \mathcal{L}}{\partial c[k \triangle j]} \overset{+}{\leftarrow} \frac{\partial \mathcal{L}}{\partial a[i \triangle j][k]}$

            **for** $k = i \ldots j - 1$ **do**

                $\frac{\partial \mathcal{L}}{\partial b[i \triangleleft j][k]} \leftarrow \frac{\partial \mathcal{L}}{\partial c[i \triangleleft j]} a[i \triangleleft j][k]$

                $\frac{\partial \mathcal{L}}{\partial a[i \triangleleft j][k]} \leftarrow \frac{\partial \mathcal{L}}{\partial c[i \triangleleft j]} b[i \triangleleft j][k]$

            $s = \sum_{k=i \ldots j-1} \frac{\partial \mathcal{L}}{\partial b[i \triangleleft j][k]} b[i \triangleleft j][k]$    $\triangleright$ Backpropagate through the softmax function

            **for** $k = i \ldots j - 1$ **do**

                $\frac{\partial \mathcal{L}}{\partial a[i \triangleleft j][k]} \leftarrow b[i \triangleleft j][k] \left( \frac{\partial \mathcal{L}}{\partial b[i \triangleleft j][k]} - s \right)$

            **for** $k = i \ldots j - 1$ **do**

                $\frac{\partial \mathcal{L}}{\partial c[i \triangle k]} \overset{+}{\leftarrow} \frac{\partial \mathcal{L}}{\partial a[i \triangleleft j][k]}$

                $\frac{\partial \mathcal{L}}{\partial c[k+1 \triangle j]} \overset{+}{\leftarrow} \frac{\partial \mathcal{L}}{\partial a[i \triangleleft j][k]}$

            **for** $k = i \ldots j - 1$ **do**

                $\frac{\partial \mathcal{L}}{\partial b[i \triangleright j][k]} \leftarrow \frac{\partial \mathcal{L}}{\partial c[i \triangleright j]} a[i \triangleright j][k]$

                $\frac{\partial \mathcal{L}}{\partial a[i \triangleright j][k]} \leftarrow \frac{\partial \mathcal{L}}{\partial c[i \triangleright j]} b[i \triangleright j][k]$

            $s = \sum_{k=i \ldots j-1} \frac{\partial \mathcal{L}}{\partial b[i \triangleright j][k]} b[i \triangleright j][k]$    $\triangleright$ Backpropagate through the softmax function

            **for** $k = i \ldots j - 1$ **do**

                $\frac{\partial \mathcal{L}}{\partial a[i \triangleright j][k]} \leftarrow b[i \triangleright j][k] \left( \frac{\partial \mathcal{L}}{\partial b[i \triangleright j][k]} - s \right)$

            **for** $k = i \ldots j - 1$ **do**

                $\frac{\partial \mathcal{L}}{\partial c[i \triangle k]} \overset{+}{\leftarrow} \frac{\partial \mathcal{L}}{\partial a[i \triangleright j][k]}$

                $\frac{\partial \mathcal{L}}{\partial c[k+1 \triangle j]} \overset{+}{\leftarrow} \frac{\partial \mathcal{L}}{\partial a[i \triangleright j][k]}$

24

---