# OpenReview forum: "Differentiable Perturb-and-Parse: Semi-Supervised Parsing with a Structured Variational Autoencoder"
_ICLR.cc/2019/Conference_

### Official Review · AnonReviewer3 · 2018-10-29
**Novel and nice method, but experiments are not strong enough**

**Rating:** 5
**Confidence:** 3

**Review:**

This paper proposed a variational autoencoder-based method for semi-supervised dependency parsing. Given an input sentence s, an LSTM-based encoder generates a sentence embedding z, and a NN of Kiperwasser & Goldberg (2016) generates a dependency structure T. Gradients over the tree encoder are approximated by (1) adding a perturbation matrix over the weight matrix and (2) relax dynamic programming-based parsing algorithm to a differentiable format. The decoder combines standard LSTM and Graph Convolutional Network to generate the input sentence from z and T. The authors evaluated the proposed method on three languages, using 10% of the original training data as labeled and the rest as unlabeled data.

Pros
1. I like the idea of this sentence->tree->sentence autoencoder for semi-supervised parsing. The authors proposed a novel and nice way to tackle key challenges in gradient computation. VAE involves marginalization over all possible dependency trees, which is computationally infeasible, and the proposed method used a Gumbel-Max trick to approximate it. The tree inference procedure involves non-differentiable structured prediction, and the authors used a peaked-softmax method to address the issue. The whole model is fully differentiable and can be thus trained end to end.

2. The direction of semi-supervised parsing is useful and promising, not only for resource-poor languages, but also for popular languages like English. A successful research on this direction could be potentially helpful for lots of future work.

Cons, and suggestions on experiments
My main concerns are around experiments. Overall I think they are not strong enough to demonstrate that this paper has sufficient contribution to semi-supervised parsing. Below are details.

1. The current version only used 10% of original training data as labeled and the rest as unlabeled data. This makes the reported numbers way below existing state-of-the-art performance. For example, the SOTA UAS on English PTB has been >95%. Ideally, the authors should be able to train a competitive supervised parser on full training data (English or other languages), and get huge amount of unlabeled data from other sources (e.g. News) to further push up the performance. The current setting makes it hard to justify how useful the proposed method could be in practice.

2. The best numbers from the proposed model is lower than baseline (Kipperwasser & Goldberg) on English, and only marginally better on Swedish. This probably means the supervised baseline is weak, and it's hard to tell if the gains from VAE will retain if applied to a stronger supervised.

3. A performance curve with different amount of labeled and unlabeled data would be useful to better understand the impact of semi-supervised learning.

4. What's the impact of perturbation? One could simply use T=Eisner(W) as approximation. Did you observe any significant benefits from sampling?

Other questions
1. What's the impact of keeping the tree constraint on dependencies during backpropagation?  Have you tried removing the tree constraint like previous work?

2. Are sentence embedding and trees generated from two separate LSTM encoders? Are there any parameter sharing between the two?

---

> ### Author Response · Authors · 2018-11-13
> **Response to AnonReviewer3**
>
> Thank you for your comments and for finding the method novel and interesting.
>
> We would like first to clarify that we are not making claiming that our method is appropriate in the high resource scenario (i.e. full in-domain English PTB parsing).  However, large datasets are available only for a few languages, so the lower resource setting we study here is important and common.  We use a sufficiently strong baseline (e.g., already using external word embeddings) and obtain improvements across all 3 languages.  Interestingly, we observe that there are certain phenomena which our semi-supervised parser captures considerably more accurately than the baseline model (e.g., long distance dependencies and multi-word expression, see reply to R1).  Very few studies have been done for semi-supervised structured prediction with  neural generative models, especially for the more challenging parsing task, so we think these results are interesting.
>
> We also think that our differentiable perturb-and-parse operator is interesting on its own, and has other potential applications.  For example, it could be used in the context of latent structure induction, where there is no supervision (i.e. no treebank). Our sampling technique has properties which are different from those of previously proposed latent induction methods:
> - unlike structured attention [4], we sample global structures rather than compute marginals (e.g., we preserve higher-order statistics)
> - unlike SPIGOT [2], we can impose tree constraints directly rather than compute an approximation
> - unlike us, [3] relies on sparse distributions so that marginalization is feasible. While sparse distributions have many interesting properties, they yield flat areas in the optimization landscape that can be difficult to escape from.
> - unlike sampling with shift-reduce parsing models,  we do not seem to have issues with bias which was argued to negatively affect its results [1].
>
>
>  > A performance curve with different amount of labeled and unlabeled data
>
> We will do our best to include these results in a subsequent revision. Using more unlabeled data is harder for Swedish and French, as we would need to re-tokenize in the form consistent with our labeled data.
>
>
> > What's the impact of perturbation?
>
> In our experiments, using sampling is beneficial so that improvements are consistent across languages. For example, UAS results in French for the model that does not us sentence embeddings are as follows:
> - supervised: 84.09
> - semi-supervised without sampling: 84.27
> - semi-supervised with sampling: 84.69
>
>
> > What's the impact of keeping the tree constraint on dependencies during backpropagation?
>
> We thought that the main motivation for dropping the constraint in previous work (e.g., SPIGOT) was efficiency.  Since it does not seriously affect computation cost in our approach, we have not experimented with dropping it.
>
>
> >  Are sentence embedding and trees generated from two separate LSTM encoders?
>
> Yes. There are no shared parameters in our model: the LSTM of the parser, the LSTM generating the sentence embeddings and the decoder are all separate. Introducing parameter sharing would likely be beneficial.  However, our set-up is more controlled, as we can make sure that the improvements are due to modeling latent syntactic structure rather than getting better word representations (i.e. from using the multi-task learning objective).
>
>
>
> [1] Andrew Drozdov and Samuel Bowman, The Coadaptation Problem when Learning How and What to Compose (2nd Workshop on Representation Learning for NLP, 2017)
> [2] Hao Peng, Sam Thomson and Noah Smith, Backpropagating through Structured Argmax using a SPIGOT (ACL 2018)
> [3] Vlad Niculae, André Martins and Claire Cardie, Towards Dynamic Computation Graphs via Sparse Latent Structure (EMNLP 2018)
> [5] Yoon Kim, Carl Denton, Luong Hoang and Alexander Rush, Structured Attention Networks (ICLR 2017)

---

### Official Review · AnonReviewer2 · 2018-11-02
**An interesting application of VAEs**

**Rating:** 7
**Confidence:** 3

**Review:**

[Summary]
This paper proposes to do semi-supervised learning , via a generative model, of an arc-factored dependency parser by using  amortized variational inference.  The parse tree is the latent variable, the parser is the encoder that maps a sentence to a distribution over parse-trees, and the decoder is a generative model that maps a parse tree to a distribution over sentences.

[Pros]
Semi-supervised learning for dependency parsing is both important and difficult and this paper presents a novel approach using variational auto-encoders. And the semi-supervised learning method in this paper gives a small but non-zero improvement over a reasonably strong baseline.

[Cons]
1. My main concern with this paper currently are the "explanations" provided in the paper which are quite hand-wavy. E.g. the authors state that using a KL term in semi-supervised learning is exactly opposite to the "low density separation assumption".  And therefore they set the KL term to be zero. One has to wonder that why is the "low density separation assumption" so critical for dependency parsing only? VAEs have been used with a prior for semi-supervised learning before, why didn't this assumption affect those models ?

A better explanation will have been that since the authors first trained the parser in a supervised fashion, therefore their inference network already represents a "good" distribution over parses, even though this distribution is specified only upto sampling but not in a mathematically closed form. Finally, setting the KL divergence between the posterior of the inference network and the prior to be zero is the same as dynamically specifying the prior to be the same as the inference network's distribution.

2. A number of important details are missing in the submitted version of the paper which the authors addressed in their reply to my public comment.

3. The current paper does not contain any comparison to self-training which is a natural baseline for this work. The authors replied to my comment saying that self-training requires a number of heuristics but it's not clear to me how much more difficult can these heuristics be than the tuning required for training their VAE.

---

> ### Author Response · Authors · 2018-11-13
> **Response to AnonReviewer2**
>
> Thank you for your suggestions and the positive feedback.
>
> > hand-wavy explanations
>
> We toned down our speculation, and incorporated your suggestions. Please let us know if you think, we could improve this further.
>
> >  A number of important details are missing in the submitted version of the paper which the authors addressed in their reply to my public comment.
>
>
> The submission has now been updated, reflecting what we described in our public comment.

---

### Official Review · AnonReviewer1 · 2018-11-05
**I thought this was an excellent paper - very clear, an important problem, a useful set of techniques and results.**

**Rating:** 8
**Confidence:** 4

**Review:**

The paper describes a VAE-based approach to semi-supervised learning
of dependency parsing. The encoder in the VAE is a neural edge-factored
parser allowing inference using Eisner's dynamic programming algorithms.
The decoder generates sentences left-to-right, at each point conditioning
on head-modifier dependencies specified by the tree. A key technical
step is to develop a method for "differentiable" sampling/parsing,
using a modification of the dynamic program, and the Gumbel-max trick.

I thought this was an excellent paper - very clear, an important
problem, a very useful set of techniques and results. I would strongly
recommend acceptance.

Some comments:

* I do wonder how well this approach would work with orders of magnitude
more unlabeled data. The amount of unlabeled data used is quite small.

* Similarly, I wonder how well the approach works as the amount of
unlabeled data is decreased (or increased, for that matter). It should
be possible to provide graphs showing this.

* Are there natural generalizations to multi-lingual data, for example
settings where supervised data is only available for languages other
than the language of interest?

* It would be interesting to see an analysis of accuracy improvements
on different dependency labels. The "root" case is in some sense just
one of the labels (nsubj, dobj, prep, etc.) that could be analyzed.

* I wonder also if this method would be particularly helpful in
domain transfer, for example from Wall Street Journal text to
Wikipedia or Web data in general. The improvements could be more
dramatic in this case - that kind of effect has been seen with
ELMO for example.

---

> ### Author Response · Authors · 2018-11-13
> **Response to AnonReviewer1**
>
> Many thanks for the positive feedback and suggestions.
>
> >  Varying amounts of unlabeled data
>
> We will do our best to include these results in a subsequent revision. Using more unlabeled data is harder for Swedish and French, as we would need to re-tokenize in the form consistent with our labeled data.
>
>
> > Are there natural generalizations to multi-lingual data  for example settings where supervised data is only available for languages other than the language of interest?
>
> This is a very interesting direction. We hope that using ‘unlabeled’ and ‘labeled’ terms in the objective would make the multilingual model capture correspondences between surface regularities and the underlying syntax, for a given language. This should be especially helpful in the suggested one-shot learning scenario, where only unlabeled term will present for the target language. We suspect that part-of-speech tags (not currently used in our model) would be needed to facilitate learning the cross-lingual correspondences.
>
>
> > I wonder also if this method would be particularly helpful in domain transfer
>
> Yes, we would like to look into this in the future work.
>
>
> > It would be interesting to see an analysis of accuracy improvementson different dependency labels.
>
> We performed analysis on English, there are some interesting cases:
> 1. Multi-word expressions: the recall / precision scores of the semi-supervised model are 90.70 / 84.78 while the one of the supervised model are 75.58 / 81.25. We suspect that the reason is that MWEs are relatively infrequent.
> 2. Adverbial modifiers: we observe an increase in precision without compromising on recall: 87.32 / 87.51 versus 87.27 / 85.95.
> 3. Appositional modifiers: we also observe a significant increase for the recall in this  category: 81.39 / 81.03 versus 77.49 / 80.27
> We included the results in the new version of the paper.

---

### Comment · AnonReviewer2 · 2018-10-31
**Clarifications**

This paper proposes to do semi-supervised learning , via a generative model, of an arc-factored dependency parser by using  amortized variational inference.  The parse tree is the latent variable, the parser is the encoder that maps a sentence to a distribution over parse-trees, and the decoder is a generative model that maps a parse tree to a distribution over sentences.  While this idea itself is exciting, a few important details are missing in the paper, that are needed to review the paper.

1. A VAE requires a generative story for the latent variables. What exactly is the distribution of p(T|n) ? This distribution is not mentioned anywhere in the paper. More importantly Section 5 focuses entirely on the first term of the ELBO objective. What about the second term, the negative KL term, of the ELBO ? How exactly do you compute KL[q_φ(T , z|s)|p(T , z)] in equation (3) ? You mention that you use a weight of 0 for the KL term during optimization in the experiments section because you did not see any benefit from the KL term ? But what was the form of the prior that you used earlier ?

2. As you mention Smith and Eisner (2008) showed how to frame dependency parsing as an MRF and Perturb and MAP is a method for sampling from the posterior for general MRFs. However, you are adding a further relaxation and replacing the argmax with a softmax operation ( where you set τ = 1 in all experiments). So at the end you no-longer get true dependency trees but continuous entries in T. How exactly do you compute  log p_θ( s | RELAXATION of Eisner(W + P) ) in this scenario ? How do you feed soft connections to the GCN ? Does T contain probabilities or log-probabilities in this case?

3. You mention a number of other fairly simple methods for semi-supervised learning such as self-training and co-training in the related work section. Clearly these types of methods will be the right baseline to evaluate against since they do not use word-embeddings, or any manual feature engineering. What was the reason to not evaluate against such simple methods?

---

> ### Author Response · Authors · 2018-11-01
> **Clarifications**
>
> Thank you for your comments and finding the idea exciting. Please find our replies to your questions.
>
> 1. Thank you for pointing this out. We experimented with a version where the prior was the uniform distribution over all projective trees. It was not effective: downweighting or remove the KL term was yielding the best results. We realize that this prior may not be quite appropriate (linguistic trees are not samples from the uniform distribution), but given that our model is generative / not conditional (e.g., we do not condition even on PoS tags), the distribution would not be sharp anyway (even if we estimate it).  This makes us sceptical about using the KL term in our semi-supervised learning: using KL with respect to a high entropy distribution forces our model to be uncertain on unlabelled sentences. This is exactly opposite of the standard “low density separation assumption”: our preference should be for models which are confident on datapoints (roughly speaking,  decision boundaries should not cross datapoints). This motivated us to try another alternative (also not yielding ELBO), where instead of the KL term we used an adversarial term forcing our model to draw trees similar to linguistic ones. Unfortunately, it was not effective as well. We would clarify this extra experiments in a new version of the paper. Note that not using ELBO should not prevent us from using the term VAE: many recent VAE versions (e.g., beta-VAE) cannot be interpreted as optimizing ELBO.
>
> 2.   Again, we should have clarified this. We rely on perturb-and-map to sample a single tree from the posterior distribution. However, the MAP procedure is not differentiable, therefore we replace it with a differentiable surrogate. In our model, the weights in T do not represent probabilities neither log-probabilities but a soft-selection of arcs. GCN can be run over weighted graphs, the message passed between nodes is simply multiplied by the continuous weights. This is actually a motivation for using GCN rather than a Recursive LSTM/RNN. On the one hand, running a GCN with a matrix that represents a soft-selection of arcs (i.e. with real values) has the same computational cost than using a standard adjacency matrix (i.e. with binary elements) if we use matrix multiplication on GPU (optimization with sparse matrix multiplication is helpful on CPU, but not always on GPU). On the other hand, a recursive network over a soft-selection of arcs requires to build a n^2 set of RNN-cells that follow the dynamic programming chart where the possible inputs of a cell are multiplied by their corresponding soft-selection in T, which is expensive and not GPU-friendly.  We also experimented with using straight-through estimators where GCN computation is performed over a discretizatized version of the graph, whereas the backpropagation step is done over the soft version. We did not see much of a difference in performance.
>
> 3.   Self-training is an option, though all (?) previous applications of self-training to syntactic parsing used quite a number of tricks and parameters (e.g., McClosky et al 2006; Reichart and Rappoport 2007,  Yu and Bohnet 2017).  Even if self-training works, we believe that our approach provides an interesting alternative, and one of very few methods for semi-supervised learning for structured prediction where improvements over a strong supervised baseline can be seen (recall that our baseline already uses external embeddings). What is also interesting is that the parse trees predicted by the semi-supervised model are qualitatively different from the ones produced by the supervised baseline. E.g., as we discuss in the experimental section, it predicts many more long distance dependencies than the supervised one. We speculate that this is an artefact of using the RNN+GCN decoder which does not care about short edges as they are too easy to encode by RNN, so encourages longer range dependencies.  This won’t happen for self-trained parsers as self-training reinforces the predictions. Co-training is even harder to make to work than self-training, as we need to come up with two models and it would be more orthogonal to our method (we could use a co-training loss in combination with ours). Previous work suggests that co-training does not work out-of-the-box for syntactic parsing, so a meaningful baseline would be hard to construct.

---

### Meta-Review · Area_Chair1 · 2018-12-13
**Novel, well-founded, and interesting method. Concerns about baseline**

**Confidence:** 4
**Recommendation:** Accept (Poster)

**Metareview:**

This paper proposes a method for unsupervised learning that uses a latent variable generative model for semi-supervised dependency parsing. The key learning method consists of making perturbations to the logits going into a parsing algorithm, to make it possible to sample within the variational auto-encoder framework. Significant gains are found through semi-supervised learning.

The largest reviewer concern was that the baselines were potentially not strong enough, as significantly better numbers have been reported in previous work, which may have a result of over-stating the perceived utility.

Overall though it seems that the reviewers appreciated the novel solution to an important problem, and in general would like to see the paper accepted.